# Photocrosslinkable Biomaterials for 3D Bioprinting: Mechanisms, Recent Advances, and Future Prospects

**DOI:** 10.3390/ijms252312567

**Published:** 2024-11-22

**Authors:** Yushang Lai, Xiong Xiao, Ziwei Huang, Hongying Duan, Liping Yang, Yuchu Yang, Chenxi Li, Li Feng

**Affiliations:** Division of Vascular Surgery, Department of General Surgery and Regenerative Medicine Research Center, West China Hospital, Sichuan University, Chengdu 610041, Chinanessen2004@163.com (X.X.);

**Keywords:** 3D bioprinting, hydrogel, photocrosslinkable biomaterials

## Abstract

Constructing scaffolds with the desired structures and functions is one of the main goals of tissue engineering. Three-dimensional (3D) bioprinting is a promising technology that enables the personalized fabrication of devices with regulated biological and mechanical characteristics similar to natural tissues/organs. To date, 3D bioprinting has been widely explored for biomedical applications like tissue engineering, drug delivery, drug screening, and in vitro disease model construction. Among different bioinks, photocrosslinkable bioinks have emerged as a powerful choice for the advanced fabrication of 3D devices, with fast crosslinking speed, high resolution, and great print fidelity. The photocrosslinkable biomaterials used for light-based 3D printing play a pivotal role in the fabrication of functional constructs. Herein, this review outlines the general 3D bioprinting approaches related to photocrosslinkable biomaterials, including extrusion-based printing, inkjet printing, stereolithography printing, and laser-assisted printing. Further, the mechanisms, advantages, and limitations of photopolymerization and photoinitiators are discussed. Next, recent advances in natural and synthetic photocrosslinkable biomaterials used for 3D bioprinting are highlighted. Finally, the challenges and future perspectives of photocrosslinkable bioinks and bioprinting approaches are envisaged.

## 1. Introduction

One of the goals of tissue engineering is to fabricate bioactive scaffolds with specific structures and functions for regenerative medicine. Researchers have developed numerous functional constructs in the past decades by combining biomaterials, seed cells, and biological molecules [1,2]. Despite the remarkable progress in biomedical applications like injury repair [3,4], disease modeling [5,6], and drug screening [7], the clinical application of tissue engineering remains unsatisfactory, which is partly attributed to the limited ability to build scaffolds with controlled cell or biomaterial distribution, interconnected vascular systems, and complex geometry [8]. For example, conventional tissue engineering has explored several strategies, such as particle leaching [9,10], gas foaming [11,12,13], solvent casting [14], and electrostatic spinning [15,16], to fabricate scaffolds with the desired porous structures, thus facilitating control over the mechanical performance [17,18], degradation properties [19,20,21,22], cellular response [23,24], and oxygen, nutrient, and metabolic waste exchange [25,26,27,28]. However, developing tissue-engineered bioactive devices with customized structures, specific cell organizations, and dynamic physicochemical niches that satisfy the personalized demands for structural and functional organ regeneration remains challenging [29,30,31,32].

To build customized scaffolds, various 3D bioprinting approaches that aim to recapitulate the key features of native organs/tissues have been developed, such as extrusion-based, inkjet, stereolithography, and laser-assisted 3D bioprinting [33,34]. Three-dimensional bioprinting is an additive biomanufacturing technique that mainly utilizes cell-laden bioinks to build biomimetic constructs in a layer-by-layer manner based on predesigned 3D models [33,34]. Through 3D bioprinting, it is possible to create scaffolds with controllable geometry, high cell density, and improved bioactivity that are suitable for regenerative medicines in vitro or even in vivo.

Despite the exciting achievements of 3D bioprinting in tissue/organ repair [35,36,37], drug delivery [38,39,40], and disease models [41,42,43], there are still several challenges that need to be taken into consideration for further research advances. One of the most important parts is the development of appropriate bioinks. Generally, bioinks comprise cells, biomaterials, growth factors, and other functional additives. As a part of bioinks, biomaterials play a crucial role in endowing them with proper printability, biodegradability, bioactivity, and physiochemical performance [44]. Hydrogel materials are the most commonly used for bioinks due to their hydrating nature being similar to native extracellular matrix (ECM).

Based on the nature of the biomaterials involved, hydrogels are formed through a range of physical or covalent crosslinking strategies, such as Schiff base reactions [45], enzymatic crosslinking [46], ionic crosslinking [47,48], hydrogen bonding [49], and photocrosslinking [50]. In particular, proper crosslinking is frequently required to guarantee scaffold fidelity and integrity in 3D bioprinting. Among biomaterials, photocrosslinkable biomaterials have attracted huge attention for their ability to rapidly fabricate 3D scaffolds with enhanced resolution and improved fidelity [51]. To date, numerous natural or synthetic photocrosslinkable biomaterials have been developed for 3D bioprinting, such as methacrylate gelatin (GelMA) [52], methacrylate hyaluronic acid (HAMA) [53], methacrylate alginate (AlgMA) [54], and Polyethylene glycol diacrylate (PEGDA) [55].

With the rapid development of light-based 3D bioprinting, it is necessary to present a general overview of the recent status and trends of photopolymerizable biomaterials within the field. In this paper, we first briefly introduce the principles, developments, and features of the most commonly photopolymerization-related 3D bioprinting technologies, including extrusion-based printing, inkjet printing, stereolithography printing, and laser-assisted printing. Then, the mechanisms and characteristics of both photopolymerization reactions and photoinitiators are discussed. Next, we focus on the recent progress and evolution of photocrosslinkable natural or synthetic biomaterials commonly used in light-based 3D bioprinting. Finally, existing limitations and future directions of photocrosslinkable materials for 3D bioprinting are discussed.

## 2. The Major 3D Bioprinting Technologies

### 2.1. Extrusion-Based 3D Bioprinting

Currently, the most popular strategies for 3D bioprinting include extrusion printing, inkjet printing, stereolithography, and light-assisted printing [33,34]. Among these 3D printing technologies, extrusion 3D printing is one of the most common [56].

In extrusion-based 3D bioprinting, the bioink is loaded into a plastic or metal barrel and then extruded from the printing needle through a piston, screw, or pneumatic extrusion to build 3D scaffolds (Figure 1a) [57,58,59]. In most extrusion-based 3D bioprinting methods, bioinks with enough viscosity in the static state are highly preferred to guarantee the stability and integrity of the printed scaffolds [60,61,62]. Meanwhile, shear-thinning properties are also required to ensure smooth extrusion, thus reducing the shear force to alleviate potential cell damage [63,64,65]. One advantage of extrusion-based 3D bioprinting is the ability to achieve rapid scaffold construction with a high cell density (10^8^–10^9^ cells/mL) [66,67]. However, because of undesirable nozzle clogging, printing failures, and high-shear-stress-induced cell death, a nozzle that is too small is not favorable for extrusion-based bioprinting. As a result, the printing accuracy is limited to around 100 µm [68,69].

### 2.2. Inkjet 3D Bioprinting

The mechanism of inkjet 3D bioprinting is similar to that of traditional office inkjet printing. Generally, low-viscosity bioink is first loaded into a cartridge before being ejected from the printing nozzle through thermal, piezoelectric, and electromagnetic forces to form tiny droplets, which are deposited on the platform to build 3D scaffolds (Figure 1b) [70,71,72]. Although utilizing piezoelectric and electromagnetic interactions is beneficial for producing smaller droplets and improving the printing resolution, it can also lead to unexcepted cell membrane damage or death, which should be taken into careful consideration in inkjet bioprinting [73,74].

For bioinks used in inkjet 3D bioprinting, low viscosity is essential for allowing the smooth formation of ink flow and droplets. In addition, quick crosslinking is beneficial for ensuring a stable printing structure. With inkjet 3D bioprinting, it is possible to obtain scaffolds with a fine resolution (10–50 µm) [75,76]. Furthermore, cells with viability higher than 90% also can be achieved because of the use of low-viscosity bioink [77]. However, the inability of inkjet 3D bioprinting to print continuously and quickly with high cell concentrations (10^6^–10^7^ cells/mL) limits its further application [78].

### 2.3. Stereolithography 3D Bioprinting

Stereolithography, first introduced in the 1980s, utilizes a photosensitive resin and a computer-controlled UV laser to build a layer of cured resin through photopolymerization in a point-by-point scanning manner. After one layer is completed, the next layer is moved back to complete the curing. Finally, the printed structure is obtained (Figure 1c) [79,80]. In recent years, stereolithography has also emerged in the field of 3D bioprinting. One of the advantages of stereolithography bioprinting is that it enables the rapid construction of biological scaffolds. Moreover, the resolution of printed scaffolds has been greatly improved due to the use of computer-guided laser scanning [81,82,83]. However, continuous high-energy UV laser scanning during printing may cause cell damage [84,85].

### 2.4. Laser-Assisted 3D Bioprinting

Laser-assisted printing was initially applied to metal transfer and later, more recently, to 3D bioprinting. In general, the printing devices consist of the laser source, target plate, and receiving substrate plate. The target plate often consists of a transparent substrate, a gold or titanium energy-absorbing layer, with bioink coated on it. When the laser emitted to the target plate is absorbed by the absorber layer, a phase change occurs, and the ink is ejected onto the receiving substrate plate (Figure 1d) [86]. The cells often exhibit high viability because few shear forces are generated during the printing process [87,88]. The computer scanning of the laser enables the construction of fine printed structures [89,90,91]. Similar to inkjet 3D bioprinting, the printing ink used for laser-assisted 3D bioprinting requires low viscosity along with secondary crosslinking to form a stable print structure.

## 3. Photocrosslinking Reaction

In addition to appropriate printing methods, printing inks play a pivotal role in modulating the scaffold’s biological and mechanical performance. The ideal bioinks should meet some basic requirements associated with printability, biocompatibility, biodegradability, and mechanical properties [92,93]. Hydrogel has attracted great attention in 3D bioprinting for its superiority in recapitulating the microenvironment features of the ECM, which is essential for cell adhesion, proliferation, and migration [94,95]. Ideally, bioinks should be viscous liquids to ensure smooth printing and then transform into hydrogels to avoid scaffold collapse.

In general, printing inks are crosslinked into hydrogels through physical and chemical interactions. Physical crosslinking mainly includes hydrogen bonding [49], hydrophobic interactions [96], electrostatic attraction [97], and ionic crosslinking [44]. The hydrogel formed by non-covalent physical interaction is mechanically weak. Chemical crosslinking utilizes various chemical strategies, such as Schiff base reactions [98], azide–alkyl cyclization [99], hydrazide–aldol coupling [100], Michael addition [101], enzymatic reactions [102,103], and UV [104,105], visible [106,107,108] and near-infrared light [109,110] crosslinking reactions, to induce covalent hydrogel crosslinking. Compared to other crosslinking strategies, photocrosslinking is suitable for 3D bioprinting for its simple, rapid, and precise control over the curing process, which helps build scaffolds with the desired structure [111,112]. In the process of photocrosslinking, the selection and concentration of the photoinitiator, intermediate products, reaction rate, and other factors are closely related to the biological activity and shape fidelity of 3D-bioprinted scaffolds. Therefore, it is necessary to understand the mechanism of photocrosslinking. Currently, there are three types of photocrosslinking reactions for 3D bioprinting: free-radical-mediated chain polymerization, thiol-ene crosslinking, and redox-based crosslinking [113,114,115].

### 3.1. Free Radical Chain-Growth Polymerization

Free radical chain-growth polymerization is the most common method for hydrogel preparation. The reaction includes three stages: initiation, propagation, and termination. In the initiation stage, the photoinitiator is decomposed under light irradiation to generate free radicals. After successful initiation, the generated radicals can react with the corresponding functional groups on the polymer backbone to produce new covalent bonds and radical intermediates. These free radical intermediates then continue to react with unreacted functional groups to generate another radical intermediate. This process continues to propagate radical species in a chain-like manner [116,117]. Propagation continues until radical quenching occurs through radical coupling (between propagating chains or between propagating chains and photoinitiator radicals) or chain transferring from propagating chains to other molecules or inhibitors (Figure 2).

Free radical polymerization reaction behaviors have been well studied in previous studies [118,119,120]. The rate of free radical polymerization is related to the monomer concentration, photoinitiator concentration, photoinitiator molar extinction coefficient, light intensity, and so on. It is obvious that increasing the monomer concentration, photoinitiator concentration, or light intensity can help to achieve faster polymerization. However, this might also result in more cell damage or death. In 3D bioprinting, these parameters should be determined carefully to balance between the curing rate and cell viability.

Several methacrylate-based bioinks (GelMA, HAMA, and PEGDA) have been extensively studied for their quick crosslinking ability and excellent biocompatibility. However, the complex polymerization kinetics make it hard to effectively predict the progress of polymerization. More importantly, since oxygen easily reacts with free radicals to form peroxyl radicals, the propagation process is prone to be interrupted or terminated. In terms of 3D bioprinting, the polymerization of the polymer backbone tends to be inhibited by ambient oxygen, which may result in delayed or failed crosslinking. As a consequence, the shape fidelity of the 3D structure could be affected [113,121]. Although oxygen inhibition is mitigated by increasing the light intensity or photoinitiator concentration, adding additives (amines, N-vinyl amines, silanes), and removing ambient O_2_, these interventions can cause undesired damage to cells during photocrosslinking.

### 3.2. Thiol-Ene Polymerization

In recent years, bioinks based on thiol-ene click chemistry have attracted increasing attention for their ability to form homogeneous hydrogels in an oxygen-tolerating and cell-friendly way. Thiol-ene click chemistry allows precise control over hydrogel properties like the crosslinking density, mesh size, and mechanical performance by changing the type, concentration, and length of the alkene group and thiol crosslinker. Typically, thiol-ene click reactions are facilitated by either catalysts or free radicals. Light-induced thiol-ene crosslinking preserved the merits of thiol-ene click chemistry while offering a powerful tool to fabricate 3D scaffolds in a temporally and spatially controlled manner.

Similar to free-radical-mediated chain-growth polymerization, thiol-ene crosslinking also requires free radicals to initiate the reaction. During the initiation stage, the free radicals first convert the thiol groups into thiyl radicals. Due to the fact that oxygen tends to abstract hydrogen from the thiol group to regenerate the thiyl radical, this process is not impeded by ambient oxygen. Then, propagation occurs when the thiyl radicals react with alkene groups to form covalent bonds and generate new radical intermediates. The radical intermediates can react with another thiol to produce a new thiyl radical or induce polymerization among alkenes (chain growth). Thus, free-radical-induced step-growth and chain-growth polymerization may co-occur in a mixed mode. The tendency to undergo step growth or chain growth mainly depends on the reactivity of the functional groups (Figure 3). For example, norbornene exclusively reacts with thiol-containing crosslinkers through step-growth polymerization. Meanwhile, methacrylates can be crosslinked via both step-growth and chain-growth polymerization.

One advantage of thiol-ene crosslinking is its ability to form a hydrogel rapidly. The reactivity of step growth is closely related to the nature of the functional groups, such as the electron density, radical intermediate stability, and steric hindrance. In work by Northrop and his coworkers on the reactivity of a series of alkene groups, the authors demonstrated that norbornene has the highest reactivity [122]. Compared with other alkenes, like acrylates, methacrylates, vinyl ethers, and vinyl esters, the inherent ring strain endows norbornene with a rich electron density that is suitable for thiyl radical attacking (Figure 3c) [122]. Based on this, bioinks utilizing the thiol–norbornene reaction have attracted enormous interest in recent research.

Another benefit of thiol-ene click chemistry is the precise control over hydrogel networks and mechanical performance. The specific reaction between certain thiol and alkene groups can be achieved by selecting the appropriate thiol or alkene, thus gaining accurate control over the hydrogel networks. The stoichiometric ratio of the thiol to the alkene is correlated with the hydrogel’s mechanical performance. Ideally, a 1:1 stoichiometric ratio would ensure the full consumption of the thiol and alkene, realizing hydrogel formation with better mechanical performance. An excess of the thiol or alkene would lead to insufficient crosslinking or weak mechanical properties. This allows the modulation of hydrogel performance in a flexible way. In addition, the remaining thiol or alkene can offer extra sites for biological modification. For example, thiol-terminated arginine–glycine–aspartic acid (RGD) peptide has been frequently conjugated with norbornene through the thiol–norbornene reaction to improve the cell adhesion performance of the hydrogels [123].

### 3.3. Redox Photocrosslinking

In redox-based crosslinking, bioinks are mainly modified with phenolic hydroxyl groups and crosslinked in the presence of a photosensitizer. Photosensitizers are molecules that can be photo-oxidized into an excited state upon light irradiation. Excited photosensitizers then produce crosslinkable phenolic hydroxyl radicals through two reactions: (i) reacting with the phenolic hydroxyl groups directly; (ii) reacting with triplet oxygen to generate reactive oxygen species (ROS) that can oxidize the phenolic hydroxyl moieties. The nearby generated phenolic hydroxyl radicals bond to form hydrogel networks (Figure 4) [124,125].

Compared with radical chain-growth polymerization, an obvious advantage of redox-based crosslinking is that ambient oxygen can accelerate phenolic hydroxyl radical production, thus promoting hydrogel formation. However, the excess ROS generated during crosslinking might result in undesired cell damage. The bioink composition should be carefully optimized to avoid potential cytotoxicity. The most common strategy to prepare redox-based bioinks is to modify the polymer chains with phenolic hydroxyl groups, such as tyramine-modified hyaluronic acid (HA-tyr) and alginate (Alg-tyr) [126,127]. In recent studies, tyrosine-rich extracellular matrix materials were also developed for 3D bioprinting [124].

### 3.4. Photoinitiator

In addition to the crosslinking strategy, the photoinitiator also plays an important role in photocrosslinking-based bioprinting. Photoinitiators are compounds that can catalyze photopolymerization by generating crosslinking agents upon light irradiation. Typically, photoinitiators are categorized as cationic and free radical photoinitiators, depending on their different initiating mechanisms. However, cationic photoinitiators are unsuitable for 3D bioprinting due to the continuous release of toxic Lewis or Broensted acid byproducts during the initiating stage [128,129].

Free radical photoinitiators are the most common photoinitiators used in 3D bioprinting. In general, free radical photoinitiators are categorized as type I and type II according to the mechanism of generating free radicals. For a type I photoinitiator, free radicals are generated directly with the cleavage of a weak bond upon light irradiation. In contrast, free radical production with a type II photoinitiator is more complex. The activated photoinitiator requires a co-initiator to serve as a hydrogen donor, thus generating secondary radicals to initiate photopolymerization (Figure 1) [130].

The photoinitiator is generally selected based on parameters such as solubility, absorption spectra, molar extinction coefficient, quantum yield, and cytotoxicity. Firstly, the photoinitiator should be water-soluble to avoid adding toxic organic solvents. Secondly, the photoinitiator should produce proper light absorption spectra that overlap well with the light sources. A high molar extinction coefficient is also preferred to guarantee sufficient quantum yield efficiency, thus ensuring fast crosslinking with safe light intensity. Furthermore, the photoinitiator should show excellent cytocompatibility at the working concentration. Given these considerations, only a few photoinitiators are suitable for 3D bioprinting.

To date, a variety of photoinitiators have been explored in 3D bioprinting (Table 1). 2-Hydroxy-1-[4-(hydroxyethoxy)phenyl]-2-methyl-1-propanone (Irgacure 2959) is a type I photoinitiator with a molar extinction coefficient of 4 M^−1^ cm^−1^ at 365 nm. Given its water-soluble nature, Irgacure 2959 allows photocrosslinking in aqueous conditions. In an early study, Williams et al. evaluated the toxicity of several UV-light-based photoinitiators, including Irgacure 2959, 1-hydroxycyclohexyl-1-phenyl ketone (Irgacure 184), and 2,2-dimethoxy-2-phenylacetophenone (Irgacure 651), in six different cells lines in detail. They demonstrated that Irgacure 2959 shows minimal cytotoxicity in various mammalian cells [131]. Despite the good cytocompatibility of Irgacure 2959, there are still some drawbacks. For example, since Irgacure 2959 only exhibits a low molar extinction coefficient (4 M^−1^ cm^−1^) at 365 nm, long-term irradiation, strong light intensity, or high photoinitiator concentration is required for efficient crosslinking. However, this may cause undesired cell damage or genetic mutations in the encapsulated cells [132]. Meanwhile, its relatively low water solubility (0.7 *w*/*v* %) has also restricted the further application of Irgacure 2959.

In recent years, another type I photoinitiator, lithium phenyl-2,4,6-trimethylbenzoyl phosphinate (LAP), has aroused more attention in light-based 3D bioprinting. Compared with Irgacure 2959, LAP’s high water solubility (8.5 *w*/*v* %) makes it suitable for various crosslinking systems [146]. In addition, in the work by Xu and coworkers, the cells within scaffolds cured by LAP showed higher cell viability than those of Irgacure 2959 [132]. LAP also exhibits much stronger light absorption (365 nm, 218 M^−1^ cm^−1^) than Irgacure 2959, which allows faster free radical generation [135]. Due to its strong initiation ability, a much lower LAP concentration is used in bioinks. Another advantage of LAP is that it can be used as a visible light photoinitiator (405 nm, 25 M^−1^ cm^−1^) to circumvent potential UV-light damage [135].

In addition to Irgacure 2959 and LAP, 2,4,6-trimethylbenzoyl-diphenylphosphine oxide (TPO) has also been explored in light-based 3D printing. TPO is a thermally stable photoinitiator that exhibits a high molar extinction coefficient at 365 nm (680 M^−1^ cm^−1^). Similar to LAP, TPO also has an absorption spectrum in the visible light range. However, since TPO is almost insoluble in water, its application in aqueous bioinks is limited. To address this problem, Pawar and coworkers prepared a new TPO nanoparticle that shows high water dispersibility and strong photoinitiating ability, which offers a solution to prepare a water-soluble TPO photoinitiator [133].

2,2′-Azobis[2-methyl-N-(2-hydroxyethyl)propionamide] (VA-086) is another water-soluble photoinitiator for light-based 3D bioprinting [147]. Compared to Irgacure 2959, VA-086 displays less cytotoxicity, even at a 10-fold higher concentration [148,149]. Upon UV light irradiation, VA-086 will dissociate to generate free radicals and N_2_ as a byproduct. Although the introduction of N_2_ will lead to an opaque appearance, the pore structure of the hydrogel is enhanced [148,150].

Although UV-light-based photoinitiators have been widely used in 3D bioprinting, the cell damage and low penetration of UV light still obstruct their further application. Due to its good cell compatibility, visible light allows the construction of hydrogels with improved biological performance. To realize visible light photopolymerization, several visible-light-based photoinitiators, such as riboflavin (RF), fluorescein (FR), camphorquinone (CQ), tris(2,2′-bipyridyl)ruthenium(II) ([Ru(II) bpy3]^2+^), and Eosin-Y, have been developed in recent years.

## 4. Biomaterials for Light-Based 3D Bioprinting

### 4.1. Natural Materials

Currently, a variety of natural and synthetic polymers are used in 3D bioprinting. The ability to mimic the physicochemical microenvironment in which the cells are located makes natural biomaterials ideal candidates for bioinks [93]. For these reasons, natural biomaterials, especially those derived from the main components of the extracellular matrix, such as collagen, gelatin, and hyaluronic acid (HA), have been widely used in 3D bioprinting [151,152,153]. In addition, numerous modification strategies have been introduced to prepare photocurable biomaterials.

#### 4.1.1. Collagen

Collagen is a class of fibrous proteins in mammalian tissues with a specific alpha-triple-helix structure that provides mechanical support and enables the adhesion and migration of cells [154]. Depending on its structure and composition, collagen is divided into three types: type I, II, and III collagen [155]. Among them, type I collagen accounts for more than 90% of the total collagen mass in the body [155,156]. Based on their excellent bioactivity, collagen hydrogels have been widely used in various applications, like skin, cartilage, and blood vessel regeneration [157,158,159,160,161].

In the presence of residual tyrosine, the photocrosslinking of unmodified collagen can be achieved by photosensitizer-mediated redox crosslinking [162,163]. Although no additional chemical modification is required, the curing process involves a long UV irradiation time [162]. To address this, collagen was modified with extra photocrosslinkable groups, like methacrylate, norbornene, and thiols, to enhance both the curing rate and mechanical strength (Figure 5) [164,165].

Although typical methacrylate collagen (ColMA) bioinks have been successfully developed in 3D bioprinting, some limitations are still worth noting. First, since collagen tends to be physically crosslinked at neutral pH and room temperature, cells are frequently dispersed within an acidic ColMA solution before pH adjustment to prepare cell-laden bioinks, which is not favorable for 3D bioprinting. Low-temperature 3D bioprinting is a potential method. In the work by Yang et al., the authors prepared low-concentration ColMA scaffolds with good biocompatibility and bioactivity by using a low-temperature DLP 3D printing technique [166]. In addition to this, ColMA also faces oxygen inhibition, ROS accumulation, and other problems [167,168,169]. To address the drawbacks of traditional ColMA, Guo et al. prepared norbornene-modified collagen (NorCol) by conjugating carbic anhydride (CA) to a collagen backbone [170]. With the extra carboxyl groups, NorCol not only exhibited improved solubility under neutral conditions but also showed excellent miscibility with alginate and gelatin. This strategy enables hybrid bioink to respond to multiple stimuli, resulting in continuous crosslinked NorCol networks in hybrid hydrogels (Figure 5) [170].

**Figure 5 ijms-25-12567-f005:**
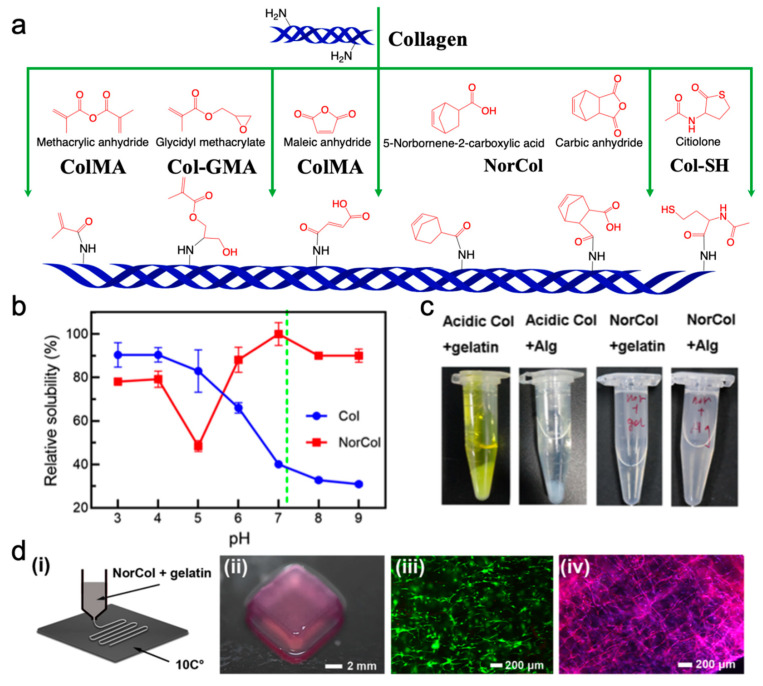
Collagen-based photocrosslinking bioinks. (**a**) Schematic representation of methacrylate- [171,172], maleic- [173], norbornene- [170], and thiol-modified [174] collagen synthesis. (**b**) The relative solubility of NorCol and collagen at different pH values. (**c**) The miscibility of NorCol with gelatin and alginate. (**d**) Temperature-sensitive extrusion bioprinting of NorCol bioinks. (**d**) (i) Schematic of temperature-sensitive extrusion bioprinting of NorCol bio-inks. (ii) Printed NorCol hydrogels (12 layers, 3 mm) after 1 day of culture. Fluorescence micrographs showing cell (iii) viability (day 1) and (iv) spreading (day 5) within NorCol hydrogels (Copyright 2021 American Chemical Society [170]).

#### 4.1.2. Gelatin

When collagen is heated to high temperatures, its triple-helix structure will irreversibly disintegrate, resulting in the formation of gelatin (Figure 6). Due to the disintegration of the supramolecular structure, gelatin has a lower molecular weight and better solubility than collagen. As the denatured product of collagen, gelatin preserves similar biological features to collagen, such as abundant RGD sequences for cell adhesion and growth [175], and retains matrix metalloproteinase (MMP)-cleavable sequences that facilitate cell migration, growth, and extracellular matrix remodeling [176,177,178]. Unlike collagen, neutral water-soluble gelatin requires no additional acid and pH adjustment operations during bioink preparation. The gelatin solution also has thermal-responsive properties but is temporarily crosslinked at low temperatures [179].

In 3D bioprinting, gelatin can serve as the sacrificial material to build constructs with vascular networks or porous structures. For permanent 3D scaffold fabrication, additional chemical modification is frequently required. Among photocrosslinkable gelatin derivatives, GelMA is the most commonly used for 3D bioprinting due to its excellent biocompatibility, good processability, and stable physical properties. In general, GelMA is synthesized through the reaction between methacrylic anhydride and the amino or hydroxyl groups of gelatin [180,181]. In the presence of photoinitiators and light irradiation, photo-reactive methacrylate groups can be activated by free radicals to form permanent covalent bonds between gelatin backbones. Due to their good bioactivity and stability, various types of cells (stem cells, progenitor cells, cancer cells, primary cells, etc.) were 3D-printed by GelMA hydrogels for tissue engineering and regenerative medicine. Despite the versatility of GelMA in cell culture and 3D bioprinting, ROS accumulation during gelation still affects cell viability [112]. In addition, cell viability is also related to the substitution degree (SD) of methacrylate groups of GelMA. He et al. prepared a series of GelMA bioinks with different methacrylate degrees and evaluated the viability of cells within GelMA scaffolds [52]. Notably, the cells embedded in GelMA scaffolds with a lower SD showed higher viability. The use of a GelMA hydrogel with low strength facilitated cell spreading in the model (Figure 7a).

To realize fast and oxygen-tolerating crosslinking, gelatin-based thiol–ene bioinks have been introduced in 3D bioprinting, such as norbornene-modified PEG (PEGNB)/thiolate gelatin (GelSH), thiolate PEG (PEGSH)/norbornene-modified gelatin (GelNB), norbornene-modified hyaluronic acid (NorHA)/GelSH, GelNB/DL-Dithiothreitol (DTT), etc. Since pure PEG-based thiol-ene bioinks lack sufficient viscosity and bioactivity, GelNB or GelSH frequently plays multiple roles in extrusion-based bioink systems: (i) a thermal-responsive ingredient for temporary crosslinking, thus ensuring scaffold integrity; (ii) a curable agent for permanent photocrosslinking; (iii) a bioactive component to facilitate cell adhesion, migration, and extracellular matrix remodeling. The synthesis of GelSH is commonly performed through the ring opening of citiolone or γ-thiobutyrolactone or the conjugation of 3,3′-dithiobis(propionohydrazide) under an inert atmosphere to avoid thiol oxidation (Figure 6).

Currently, the synthesis routes for GelNB mainly include 5-norbornene-2-carboxylic acid and CA-based modification (Figure 7f). The reaction between 5-norbornene-2-carboxylic acid and gelatin is cumbersome and requires multiple steps to synthesize GelNB [180,182]. In contrast, the direct reaction between CA and gelatin was able to achieve GelNB with a substitution degree of about 44% [183]. However, excess water-insoluble CA was used during the reaction to ensure effective grafting. This may affect the purity of the final product. Alternatively, a dual-solvent system composed of DMF/water can achieve the synthesis of GelNB with a degree of substitution of about 90.1% [184].

Although GelNB/DTT is a versatile and powerful material for 3D bioprinting, bioinks based on GelNB and thiolate macromolecular crosslinkers have been developed to improve the curing speed and avoid the potential toxicity of DTT. Thiolate macromolecular crosslinkers also protect the cell from excess ROS damage. In the work by Zhao et al., who introduced thiolate heparin (HepSH) as a macromolecular crosslinker, lower ROS levels in the GelNB/HepSH hydrogel-loaded human umbilical vein endothelial cells (HUVECs) were found [112]. Compared with GelMA, HUVECs within a 3D-printed GelNB/HepSH hydrogel exhibit better viability and spreading ability (Figure 7b,c) [112]. As a cytotoxic molecule, the uncrosslinked traditional bifunctional thiol crosslinker DTT can lead to undesired cell damage during long-term storage or printing operations. This toxicity was greatly avoided by using a macromolecular crosslinker. In another work by Göckler et al., this cytotoxicity was greatly avoided by introducing GelSH as a macromolecular crosslinker [180]. In addition, GelNB/GelSH bioinks were able to undergo superfast gelation (1–2 s), even though the photoinitiator concentration was reduced dramatically (0.03%), and promoted continuous cell growth (Figure 7d,e) [180]. Hence, the thiol-ene-based gelatin system is a promising candidate to replace GelMA. 

#### 4.1.3. Hyaluronic Acid

Hyaluronic acid (HA) is an extracellular matrix-derived glycosaminoglycan with repeat units consisting of D-glucuronic acid and N-acetylglucosamine [185,186,187]. The abundant carboxyl and hydroxyl groups endow HA with superior hydrophilicity and provide pendent sites for chemical modification. Due to its excellent biocompatibility and biodegradability, HA has been widely used for applications such as drug delivery, cell encapsulation, and wound dressings [105,188,189,190,191,192]. However, the pure HA solution is unable to form a stable hydrogel for cell encapsulation. In general, extra modification is required to endow HA with light-curing ability, such as alkenyl modification, thiolation, and tyrosine functionalization (Figure 8a).

Similar to gelatin, methacrylate HA (HAMA) is synthesized by grafting methacrylic anhydride onto the HA backbone via an esterification reaction [193,194,195]. To synthesize NorHA, HA is usually converted into intermediates like HA-TBA or HA-ADH, followed by amidation in the presence of a catalyst [196,197,198,199,200,201,202]. The synthetic route involves cumbersome and complex steps (Figure 8d). In addition, the NorHA hydrogel may become hydrophobic due to the consumption of hydrophilic carboxyl groups. More importantly, relevant studies have shown that the depletion of carboxyl groups also weakens the CD44 binding ability [203]. Alternatively, direct functionalization via esterification between CA and hydroxyl groups is a good choice to avoid the cumbersome synthesis step and the depletion of carboxyl groups (Figure 8a) [184,204]. In the work by Galarraga et al., the authors synthesized norbornene-modified HA (NorHA_CA_) based on CA and found that the extra carboxylic group enables the accelerated degradation of thiol-ene hydrogels (Figure 8b,c) [204]. When combined with stable NorHA, the NorHA_CA_/NorHA hydrogel exhibits a tunable degradation profile. NorHA_CA_/NorHA showed great promise in fabricating hydrolytically degradable bioactive scaffolds through DLP. 

**Figure 8 ijms-25-12567-f008:**
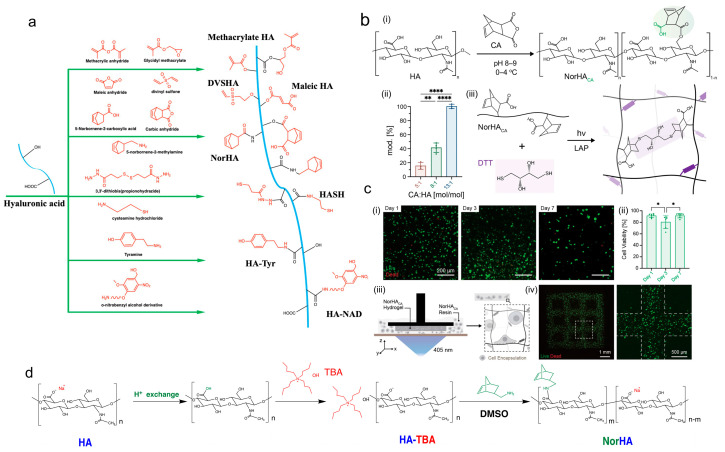
Hyaluronic acid-based photocrosslinking bioinks. (**a**) Synthesis route of light-cured hyaluronic acid. (**b**) Modification of sodium HA with CA to form NorHA_CA_. (i) Reaction scheme for NorHA_CA_ synthesis. (ii) Degree of modification of HA with norbornene is tuned by changing the molar ratio of CA to HA repeat units. (iii) Schematic representation of network formation by visible light-induced thiol-ene step-growth reaction between NorHA_CA_ and DTT in the presence of photoinitiator (LAP), (*** p <* 0.01, ***** p <* 0.0001). (**c**) Biocompatibility and DLP-based 3D bioprinting of NorHA_CA_ bioinks. (i) Representative fluorescence micrographs of bMSCs encapsulated in NorHA_CA_ (5wt%, 40%mod.) bulk hydrogels over time (1, 3, and 7 days), scale bars = 200 μm. (ii) Semiquantitative analysis of cell viability, (** p <* 0.05). (iii) Schematic representation of DLP-based 3D printing of NorHACA hydrogels with bMSCs. (iv) Representative maximum projection image of bMSCs encapsulated in a NorHA_CA_ macroporous lattice at day 1, scale bars = 1 mm and 500 μm (Copyright 2023 American Chemical Society [204]). (**d**) HA-TBA mediates the synthesis of NorHA.

For extrusion-based 3D bioprinting, which is the most common among 3D bioprinting methods, pure hyaluronic acid solutions cannot form self-supporting filaments [153]. Therefore, proper pre-solidification is frequently introduced for extrusion-based 3D bioprinting of HA. Meanwhile, combining HA with other natural or synthetic polymers, such as gelatin [181,189,195], collagen [205], methylcellulose [193,206], sodium alginate [207,208,209], polyethylene glycol [210,211], PF127 [212], or polycaprolactone [190,213], is another strategy to achieve scaffold building.

#### 4.1.4. Alginate

Alginate is a brown-algae-derived polysaccharide macromolecule consisting of β-D-mannuronic acid (M) and α-L-guluronic acid (G) repeats (Figure 9a). When multivalent cations M^n+^, such as Ca^2+^, Mg^2+^, and Ba^2+^, are inserted into the negatively charged polysaccharide chains of alginate, an egg structure is formed through ionic attractions, which results in a rapid sol–gel transition (Figure 9a) [214]. Since alginate can be cured mildly at normal temperature and pH, various cells, such as Schwann cells [47], BMSCs [215], fibroblasts [216], and HUVECs [217], have been 3D-printed with alginate bioinks.

Despite mammals lacking enzymes that can degrade sodium alginate, ionically crosslinked alginate scaffolds are still subject to degradation by ion exchange in vivo [220,221]. Apart from this, the impact of ionic crosslinkers on the viability of loaded cells is another question in alginate-based 3D bioprinting. As excess multivalent cations may cause undesired cell damage, the concentration of ionic crosslinkers should be carefully selected. Therefore, it is necessary to endow alginate with a light-curing ability to improve stability and biocompatibility. Photocurable AlgMA also can be prepared via methacrylic anhydride [214,222]. Compared to a purely ionically crosslinked alginate hydrogel, the structural stability of covalently crosslinked AlgMA scaffolds was significantly improved [214].

Generally, norbornene-modified alginate (AlgNB) can be synthesized through amidation between 5-norbornene-2-methyamine and carboxyl groups. Although the hydrophilicity of the thiol-ene hydrogel might be affected, this provides extra sites for bioactive modification. In particular, due to its non-fouling nature, the pure alginate hydrogel lacks a suitable environment to facilitate cell adhesion, migration, and growing [223]. Numerous studies have shown that the bioactivity of alginate can be improved by mixing it with hyaluronic acid [208], collagen [224], or gelatin [225] to form a composite bioink. In addition, the cell adhesion capacity of sodium alginate can be enhanced by RGD grafting of sodium alginate [218,226]. In a typical work, Ooi et al. created a thiol-ene crosslinking Alg-norb hydrogel that exhibited tunable mechanical and swelling properties, rapid gelation, and excellent bioactivity for extrusion bioprinting (Figure 9b) [212]. The thiol RGD sequence (CGGGRGDS) was grafted on Alg-norb through a photoinitiated thiol-ene reaction. Similarly, in another work by Liu et al., the RGD sequence (CGDS) was conjugated to maleimide-modified alginate (Alg-Mal) through a click reaction (Figure 9c) [219]. Although this requires a longer reaction time, these printed scaffolds exhibit improved biological performance.

#### 4.1.5. Silk Fibroin

Silk fibroin (SF) is a natural material derived from the Bombyx mori silkworm and is composed of an H-chain (350 kDa), L-chain (26 kDa), and amorphous glycoprotein P25 (30 kDa) with a ratio of 6:6:1 (Figure 10a) [227,228]. Considering its excellent biocompatibility, stable mechanical properties, biodegradability, and low cytotoxicity, it is no surprise that the FDA approved SF for biomedical applications. To date, SF has been selected for various aspects of regenerative medicine, including bone regeneration [229], cartilage repair [230], wound healing [231], and angiogenesis [231,232]. For example, it has been demonstrated that silk fibroin can promote the differentiation of mesenchymal stem cells toward an osteogenic phenotype, which makes SF an ideal choice for bone tissue engineering [233,234].

Depending on its great bioactivity, SF is also a promising candidate to serve as a scaffold backbone for 3D bioprinting. Due to the existence of strong hydrophobic interactions and H-bonding, the SF hydrogel is more stable and stronger than other natural biomaterials [235]. However, the printability of pure SF bioinks is still limited by low viscosity. Hence, SF is frequently blended with other biomaterials to reach proper viscosity, thus improving the structural integrity and stability of the printed scaffolds [236]. For example, in the work by Moon and coworkers, SF was blended with iota-carrageenan (CG). The prepared SF/CG ink exhibited suitable viscosity and shear-thinning properties, which ensures high shape fidelity (Figure 10b) [237]. However, to build stable constructs, extra light irradiation is required to initiate redox-mediated dityrosine crosslinking (Figure 10b). Thanks to its natural tyrosine residues, which endow SF with an inherent redox-based photocrosslinking ability, no additional chemical conjugation of photo-reactive moieties is required [238].

Although it has been reported that SF bioinks can complete gelation within seconds under visible light irradiation, the influence of oxidated photoinitiators on the survival of embedded cells remains challenging [239,240]. To this end, photocrosslinkable SF derivatives such as norbornene-modified SF (SF-NB) [241] and methacrylate SF (SFMA) [242] were synthesized. Through photocrosslinking modification, it is possible to realize cell-friendly crosslinking and enhance physical performance. Bhar et al. prepared a composite bioink containing SFMA, GelMA, and photoactivated human platelet releasate (PPR) to build an immunocompetent human skin model, where SFMA was employed to maintain the strength, stability, and elasticity of the printed constructs (Figure 10c) [243]. With this design, the printed artificial model showed tunable physical properties to support the biomimetic skin epidermal and dermal structures. 

**Figure 10 ijms-25-12567-f010:**
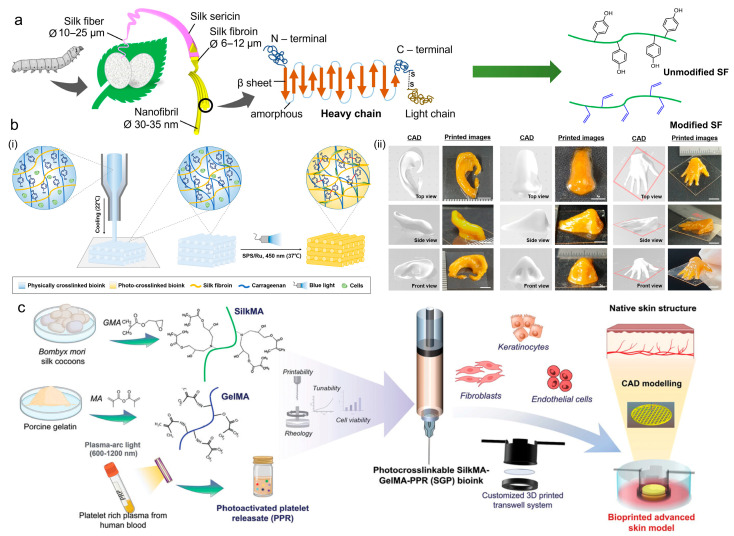
SF-based photocrosslinkable bioink. (**a**) Production, structure, and modification of SF (Copyright 2023 Elsevier [228]). (**b**) The design and printing performance of the redox-crosslinkable SF/CG bioink. (i) Schematic of the printing process of SF/CG bioink. (ii) CAD images depicting the ear, nose, and hand and printed images at various angles (Copyright 2024 Elsevier [237]). (**c**) Methacrylate-group-functionalized SF for recapitulating human skin models through 3D bioprinting (Copyright 2024 Wiely [243]).

#### 4.1.6. Decellularized Extracellular Matrix

A decellularized extracellular matrix (dECM) is a natural tissue- or organ-derived material obtained by chemical or physical decellularization techniques (Figure 11a). Apart from its great biocompatibility and biodegradability, dECM is superior in preserving the in vivo physical, mechanical, and biological microenvironment, thus facilitating tissue-specific cell proliferation and differentiation. Currently, dECM bioinks are extensively developed for cartilage [244,245,246], heart [247], liver [248,249], skin [250], and tendon regeneration [251] for their ability to offer tissue-specific cues to guide cell growth.

The viscous dECM solution is thermally responsive and experiences invertible sol–gel transition at body temperature, which makes dECM suitable for extrusion-based 3D bioprinting. However, due to unstable physical gelation, as-printed dECM scaffolds are often mechanically weak with unsatisfactory resolutions. Therefore, mechanical reinforcement strategies, such as incorporating support materials and extra crosslinking, are frequently introduced to improve printability and integrity. In addition to co-printing with synthetic or natural supporting materials (e.g., PCL, gelatin, alginate) and adding crosslinkers (e.g., genipin), photocrosslinking modification is another common approach to stabilizing dECM 3D scaffolds.

Moreover, a suitable mechanical clue is critical for cells to differentiate and function in an organ-specific way. Through proper photocrosslinking, it is possible to build a dECM hydrogel with enhanced and tunable mechanical strength similar to the original tissue or organ. Like silk fibroin, many dECM materials are readily available for redox photocrosslinking due to their abundant native tyrosine residues. One advantage of this approach is the effective protection of pristine bioactivity. In a typical work, Jang et al. realized the photocrosslinking of dECM when vitamin B2 was used as a UV-light-sensitive photoinitiator [256]. Although vitamin B2 is approved by the FDA for the treatment of corneal disorders, the use of UV light can lead to potential DNA damage. For the application of such dECM bioinks, visible light is certainly a more attractive option to initiate crosslinking. To address these issues, Ru/SPS was developed to realize safe and efficient visible-light-mediated crosslinking of dECM (Figure 11b) [253].

For 3D bioprinting, Kim and coworkers reported porcine heart (hdECM) and bovine eyeball (Co-dECM) dECM-derived bioinks, which can be crosslinked rapidly within 5 s through Ru/SPS-mediated visible light crosslinking. The printed scaffolds exhibit improved printability, physical performance, and excellent bioactivity at the centimeter scale (Figure 11c) [124]. However, the potential damage induced by excess oxidated photoinitiator is still the main obstacle to the further application of dECM bioinks.

Chemically modified photo-reactive dECM (e.g., dECM-MA, dECM-NB) allows controllable rapid gelation and enhances mechanical stability while avoiding the use of oxidative photoinitiators [257]. Similarly, methacrylation is one of the main methods to modify dECM with photo-reactive ability, which mainly includes glycidyl methacrylate (GMA) or methacrylic anhydride (MA) functionalization. Although the structural integrity of dECM is not strongly affected by GMA or MA modification, more methacrylate groups can be grafted on dECM through a ring-opening reaction between GMA and amine or hydroxyl groups (Figure 11d). In addition, with the introduction of additional hydrophilic hydroxyl moieties, the GMA dECM hydrogel exhibited better bioactivity [254]. Similarly, in another work by Duong et al., CA was used to create norbornene-functionalized small intestine submucosa (dSIS-NB), which enabled the introduction of extra carboxyl groups (Figure 11e) [255]. The dSIS-NB bioinks showed versatility in building biomimetic scaffolds to support fast cancer cell spreading and angiogenesis through both DLP and extrusion bioprinting [255].

### 4.2. Synthetic Materials

#### 4.2.1. Polyethylene Glycol

In contrast to natural biomaterials, synthetic polymers can be prepared in large quantities with controllable molecular weights and functional groups. Polyethylene glycol (PEG) is a linear polymer synthesized from ethylene glycol via polymerization. PEG has been approved by the FDA for pharmaceutical and medical applications due to its excellent biocompatibility and low immunogenicity [258]. It is no surprise that PEG and its derivatives have been widely explored in photocrosslinkable bioinks to serve as polymer backbones, crosslinkers, or sacrificial materials [259,260,261]. With terminal hydroxy groups, PEG can react with methacryloyl chloride or acryloyl chloride under basic conditions to synthesize PEGMA or PEGDA through esterification. The PEGMA or PEGDA precursor is readily crosslinked into hydrogels through chain-growth polymerization [55,262]. The physical properties of the PEG hydrogel can be efficiently adjusted by varying the molecular weight and monomer concentration. For thiol-ene-based photocrosslinking, PEG is frequently thiolated or norbornene-functionalized into various derivatives such as PEG-4SH and PEG-4NB, thus realizing versatile control over the hydrogel properties (Figure 12a) [263,264,265].

Providing a proper microenvironment to promote cell growth is essential for bioinks. Despite its good biocompatibility and hydrophilicity, it is hard for PEG to support continuous cell growth due to the lack of suitable cell adhesion and enzymatically degradable sites. Therefore, a variety of natural biomaterials have been blended with PEG to improve its bioactivity, such as gelatin, collagen, and hyaluronic acid [211,269,270]. Meanwhile, the bioactivity of PEG-based bioinks can also be effectively improved by incorporating biological ingredients like RGD sequences or heparin-binding sites [271,272,273,274]. For instance, Schwegler et al. incorporated a chemically synthesized RGD sequence into PEGDA, thereby giving the printed material cell adhesion sites without changing its mechanical properties (Figure 12b) [266]. The most common strategy to address the lack of MMP-degradable sites is introducing MMP-responsive crosslinking agents [275,276]. For instance, Chen et al. introduced a tunable-degradability hydrogel, which was synthesized by integrating MMP-sensitive peptides with norbornene-modified eight-arm polyethylene glycol macromers [277]. To endow the PEG hydrogel with dynamic mechanical performance similar to matrix remodeling, Wiley et al. developed a dual-enzyme (thrombin and MMP)-degradable peptide linker. The obtained PEG hydrogel allows a dynamic modulation process similar to matrix remodeling (Figure 12c) [267].

In another work, Zengin et al. developed a composite bioink containing MMP-sensitive peptide-modified mesoporous silica nanoparticles (MSN-MMPs), which resulted in an MMP-9-biodegradable hydrogel (Figure 12d) [268]. Furthermore, additional cysteine-modified RGD peptide incorporation enhanced cell-matrix interactions and supported the viability and proliferation of MG63 cells within 3D-bioprinted scaffolds (Figure 12d) [268].

#### 4.2.2. Pluronic F127

Pluronic is a class of copolymers consisting of hydrophilic polyethylene oxide (PEO) and hydrophobic polypropylene oxide (PPO). Pluronic F127 (PF127) is a triblock polyether copolymer of PEO-PPO-PEO (Figure 13a). When the temperature is raised to the lowest critical solution temperature (LCST), the hydrophobic chains of PF127 will aggregate. In contrast, the hydrophilic chains will stretch into the aqueous phase to form microspheres. Finally, a hydrogel is formed through the aggregation of microspheres (Figure 13a) [278]. Based on its innate temperature-sensitive property, PF127 is widely used in 3D bioprinting to act as a sacrificial ingredient, polymer backbone, or supporting material [279,280,281]. Zheng et al. reported an electrohydrodynamic (EHD) inkjet printing system composed of GelMA and PF127, where PF127 served as a sacrificial matrix to create biomimetic microvascular structures (Figure 13b) [282]. Human dermal fibroblasts (HDFs) and HUVECs were successfully co-cultured to form a structure with tissue-specific morphology and high spatial resolution (30 μm). In another work by Lewis et al., PF127 was co-printed with liver dECM to fabricate biomimetic geometry to guide the directional formation of biliary trees. 

Although PF127 can form a hydrogel at body temperature, a pure PF127 hydrogel tends to disintegrate rapidly upon the penetration of ambient water, which is unacceptable for permanent 3D device construction. Therefore, the modification of PF127 is frequently performed, like esterification between acryloyl chloride and hydroxy groups. Similar to PEGDA, the stability and fidelity of 3D-printed PF127 diacrylate (PF127DA) scaffolds are greatly improved by covalent crosslinking [285,286,287]. Since PF127 lacks cell adhesion sites, it is often co-printed with natural polymeric materials to improve bioactivity [280]. In the work by Millik et al., they reported a coaxial-nozzle-mediated tubular coextrusion printing system, where the pure PF127 of the inner phase served as a sacrificial material to guide tube formation. At the same time, the F127–bisurethane methacrylate (F127-BUM) of the outer layer was photocrosslinked to create permanent constructs [288]. After functionalization with collagen I to promote cell adhesion, it is possible to build luminal scaffolds (∼150 μm) with monolayers of HUVECs.

#### 4.2.3. Polyvinyl Alcohol

Polyvinyl alcohol (PVA) is a nontoxic, biodegradable, and biocompatible hydrophilic linear polymer that has been approved by the FDA for biomedical applications. Similar to PF127, PVA has also served as a sacrificial material [289], supporting agent [290], and hydrogel backbone [291]. One outstanding feature of PVA is its abundant hydroxyl groups, which allow physical crosslinking through H-bonding. Although toxic additives are not required for physical crosslinking, the PVA hydrogel prepared through H-bonding requires long, repeated freeze–thaw cycles, which is impractical for cell encapsulation [292]. In contrast, PVA can be crosslinked through chemical crosslinkers (boric acid, aldehydes, or epichlorohydrin) under mild conditions, which makes it possible to realize stable cell encapsulation [293]. However, the toxicity of small chemical crosslinking molecules is still one of the main obstacles to further biological advances. Therefore, it is necessary to perform extra chemical modifications to realize safe cell encapsulation and 3D bioprinting of PVA. The abundant hydroxyl groups provide enough sites for chemical modification, such as amine [294], carboxyl [295], methacrylate [296], norbornene [294], and thiol [297] groups.

Generally, PVA is often modified by photo-reactive moieties like methacrylate or norbornene to create a photocrosslinked hydrogel with stable and strong networks. The improved gelation speed, enhanced stability, and preserved biocompatibility have made the photocrosslinked PVA hydrogel a promising candidate for tissue engineering and regenerative medicine. For example, Guo and coworkers developed a composite hydrogel composed of methacrylate-modified PVA (PVAMA), AlgMA, and GelMA to induce pancreatic differentiation of induced pluripotent cells (iPSCs) [298]. The incorporation of PVAMA and AlgMA provided a high-water-content environment to support cell migration [298].

Meanwhile, PVA-based photocrosslinkable bioink also plays an important role in 3D bioprinting. In a study by Lim and coworkers, the authors developed a PVAMA and GelMA composite bioink that enables high resolution for DLP-based bioprinting (25–50 μm) [299]. The printed constructs were able to support the growth and maintain the stemness of encapsulated stem cells [299]. Based on the merits of thiol-ene chemistry, norbornene-modified PVA (nPVA) was also synthesized through esterification between CA and hydroxyl groups. Zhu and coworkers prepared a PVA-NB hydrogel through thiol-ene photocrosslinking, which showed great potential to serve as a tissue scaffold for pelvic organ prolapse treatment [300]. In another work, Qiu and coworkers developed a dynamic bioink based on nPVA and gelatin, which allows the rapid fabrication of cell-laden constructs through tomographic VBP (Figure 13c) [284]. The fabricated constructs exhibit controlled physicochemical performance that facilitates continual cell growth, supports the osteogenic differentiation of stem cells, and realizes aligned multicellular aggregates [284]. Although pure photocrosslinkable PVA can be 3D-printed into scaffolds with desired mechanical properties and fidelities, PVA is frequently functionalized by RGD or blended with natural biomaterials to improve the bioactivity due to its antifouling nature and lack of cell adhesion sites [299,301]. For summary, the Representative biomaterials for light-based 3D bioprinting were presented at Table 2.

### 4.3. Biocompatibility, Degradation, and Applications

Biocompatibility means that biological materials and their degradation products are nontoxic to cells in vitro and can perform their functions in vivo without causing adverse reactions [321]. Biocompatibility testing methods are generally divided into cytotoxicity assessment in vitro and systemic toxicity assessment in vivo. For example, experiments in vitro often treat cells with the extracts or solutions of biomaterials to evaluate their impact on cell viability, growth, proliferation, and migration. Experiments in vivo typically implant biomaterials under the skin or in the muscle tissue of rodents to see whether there are potential foreign body reactions, toxicity, or carcinogenic effects [114]. The biomaterial used for 3D bioprinting should be selected on the basis of both good printability and great biocompatibility. Despite the choice of biocompatible natural or synthetic biomaterials, careful optimization of the printing system to avoid the side effects induced by shear forces, piezoelectric interaction, light irradiation, or the photoinitiator during light-based 3D bioprinting is required. For example, the light source should be carefully selected since short-wavelength UV light might lead to serious DNA injury, and NIR light can result in severe thermal damage [322].

With the increasing application of biomaterials in the biomedical field, it is critical to understand the degradation rates and mechanisms of biomaterials to determine their applicability. Biodegradable biomaterials are usually degraded by hydrolysis or proteolysis [323]. Proteolysis requires that implanted biomaterials have biodegradable peptide sequences that can be recognized and cut off by enzymes produced by cells in or around the biomaterials, such as collagen and gelatin, which all have matrix metalloproteinases [324]. The advantage of proteolysis is that the rate of degradation is closer to the rate of cell growth. Another method of degradation is hydrolysis, which is much slower and applicable to alginate. This is beneficial if a longer-lasting implant is desired.

Hard tissue refers to tissue that forms in the body through biomineralization, such as bone tissue. Natural biomaterials such as collagen and gelatin have been widely used in the 3D printing of bone tissue due to their good biodegradability and biocompatibility [325]. However, due to their poor mechanical properties, they are often mixed with synthetic polymer materials. In addition to mechanical properties, the printed bone tissue should also be bone conductive, so biomolecules such as bone morphogenetic proteins and growth factors are often added to bioinks to induce bone signaling [326].

The 3D printing of soft tissues such as heart tissue also has specific requirements for bioinks. Blood vessel formation is an issue that must be considered in the 3D printing process due to the high blood vessel density of heart tissue. Sacrificial inks such as gelatin and Pluronic are often used to form hollow channels with smaller diameters [327]. In addition, in order to achieve the specific functionalization of the bioink, the bioink formula also needs to be adjusted accordingly; for example, the incorporation of extracellular matrix proteins like collagen and connexin into the bioink can promote cell adhesion and proliferation and matrix remodeling. Secreted small molecules such as transforming growth factors can also promote the maturation of printed heart tissue [328].

## 5. Conclusions and Future Perspectives

This paper provides an overview of photocrosslinking 3D bioprinting methods and light-curable biomaterials for photocrosslinking. We review the printing principles and current limitations of extrusion, inkjet, stereolithography, and light-assisted bioprinting and summarize the three photocrosslinking reaction mechanisms, photoinitiator types, and the advantages, disadvantages, and current applications of various photocured biomaterials. Among the cutting-edge 3D bioprinting methods, photocrosslinking-based 3D bioprinting allows the rapid, facile, and precise fabrication of bioactive scaffolds. During the printing process, the 3D structure and physical properties of the printed material can be controlled by adjusting the light duration, light intensity, and photoinitiator concentration to meet requirements. Despite the rapid development of photocrosslinking-based 3D bioprinting, there are still some challenges.

### 5.1. Cell Survival

It is important to ensure high cell viability in the bioink during the printing process. Shear stress is one of the reasons for the decrease in cell viability. No matter which printing method is used, the bioink will inevitably produce certain shear stress. However, high shear stress can deform cells and destroy cell membranes, resulting in reduced cell viability. The extrusion needle diameter and shape, the printing speed, and other printing parameters will affect the shear stress. For example, although a smaller-diameter extrusion needle has the advantage of improving printing accuracy, the increased shear stress and extrusion stress will affect cell viability. Therefore, it is necessary to evaluate the effect of different printing parameters on cell viability when printing. Meanwhile, exploring approaches that alleviate undesired cell damage will improve the survival of cells and enhance their viability. Microencapsulation and nanoencapsulation are expected to be potential methods. Their advantage is that they can envelope the cells inside to reduce the stimulation of the external environment on the cells.

In addition, the fabrication of large functional constructs for tissue engineering is another issue for 3D bioprinting. Properly interconnected vascular networks are key to the transportation of nutrients, oxygen, and metabolic waste. Through embedded printing and sacrificial printing, it is possible to build large constructs with enhanced nutrient supply [329]. Control over the capillary-like structure remains challenging. Despite DLP-based 3D bioprinting offering a powerful tool to address this, the advanced fabrication of large constructs with perfusive microchannels is difficult.

### 5.2. Development of Crosslinking System

The mechanical injury induced by mechanical parameters such as electronic, thermal, light, and shear forces during printing is still one of the reasons for low cell viability. Exploring approaches that alleviate undesired cell damage will improve the survival of cells and enhance their viability. In light-based 3D bioprinting, one of the main challenges is cell damage induced by light, especially high-energy, low-wavelength UV light. Besides its convenience and precision merits, excess light irradiation has also proven detrimental to cell survival, growth, and function. Hence, it is necessary to explore bioink systems capable of ultrafast gelation under cell-compatible light irradiation. This can be achieved by developing a highly light-reactive polymer backbone, screening novel, efficient photoinitiators, or adding light-protective UV absorbers. The balance between the photoinitiator’s exciting efficiency and the light source should be carefully considered to minimize cell damage. There is a need to develop biosafety photoinitiators with high molar extinction coefficients in safer light regions, thus achieving cell-compatible photocrosslinking.

Apart from light irradiation, the photoinitiator is another factor that strongly affects the viability and function of the embedded cells. This might mainly be attributed to the inherent toxicity of the photoinitiator, as well as the generation of cell-damaging free radicals and other byproducts. Therefore, it is necessary to develop a photoinitiator-free photocrosslinking strategy for 3D bioprinting. Several photoinitiator-free hydrogels based on crosslinking mediated by bismaleimides [330], acrylate [331], coumarin derivatives [332], o-nitrobenzene derivatives [333], azide [334,335], and styrylpyridinium (SbQ) [336] have been successfully created (Figure 14). However, most hydrogels were crosslinked upon long and strong UV irradiation, which is a challenge in cell-laden 3D bioprinting. The photoinitiator-free crosslinked hydrogel is still mainly used to fabricate cell-free constructs.

### 5.3. In Vivo Bioprinting

Another issue in photocrosslinking 3D bioprinting for building constructs in vivo is limited light penetration, which restricts the method’s feasibility. Due to the weak penetration of UV or visible light, which are the most commonly used, the gelation of bioinks within the deep region of large constructs is difficult, which can result in undesired defects. In addition, the poor tissue penetration ability also limits the in vivo application of UV or visible light. With a strong tissue penetration ability, near-infrared (NIR) light has been reported to hold great potential in realizing in vivo 3D bioprinting (Figure 15) [338,339]. However, the thermal effect of NIR light should be carefully considered in in vivo applications.

### 5.4. Clinical Conversion

Biological scaffolds composed of biomaterials and living cells have complex mechanisms and unknown long-term effects. In the 3D printing process, a small change in multi-component biomaterials can cause significant and unpredictable changes [341]. At present, there are no established regulatory standards for 3D printing technology, biomaterials, or the entire printing process. Therefore, in order to achieve the transformation of biological scaffolds from the laboratory to the clinic, the physical and chemical properties, biocompatibility, degradation performance, and biological activity of biological scaffolds in vivo should be rigorously evaluated in a variety of large animal models before clinical application to meet clinical needs [342]. In addition, in order to more quickly and effectively translate this process to the clinic, there is an urgent need to develop strict regulatory and safety guidelines to evaluate the toxicity of biomaterials and incorporated biomacromolecules such as photoinitiators, as well as the therapeutic effects and potential adverse effects of biological scaffolds [343].

To summarize, although photocrosslinking 3D bioprinting remains challenging in many ways, photocrosslinkable hydrogels remain one of the most promising materials in tissue engineering. It is believed that with progress in biomaterials and tissue engineering, photocrosslinking 3D bioprinting will be more widely used.

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
