# Peer review of "Photocrosslinkable Biomaterials for 3D Bioprinting: Mechanisms, Recent Advances, and Future Prospects"

_ijms, 2024, doi:10.3390/ijms252312567_

Round 1
Reviewer 1 Report
Comments and Suggestions for Authors
In this manuscript, the authors presented a review of photopolymerization-related 3D bioprinting approaches, including a discussion of the topics related to extrusion-based printing, inkjet printing, stereolithography printing, and laser-assisted printing. Moreover, the mechanism, advantages, and limitations of photopolymerization and photoinitiators are reviewed. This is a meaningful work for the design of photocrosslinkable biomaterials and I would like to recommend its publication in International Journal of Molecular Sciences after the following issues are addressed:
1. Compared to other crosslinking methods, the authors should explain the advantages of photocrosslinking reactions in 3D bioprinting.
2. In “The major 3D bioprinting technologies” part, Figure 1 consists of four parts. The related descriptions in the text are ambiguous and need to be stated more clearly. Such as Figure 1a, Figure 1b…
3. For biomaterials, Low-temperature 3D printing technology should be mentioned.
4. Faced with the dilemma that most 3D bioprinting is still in the laboratory stage, there is an urgent need for printing technologies that can be put into practical applications on the market to prove that bioprinting is not just a "paper talk". In fact, many of those experimental technologies still rely on extrusion-based methods, which are usually functional, but when it comes to printing tissues that heavily rely on arrangement, such as muscles and nerve fibers, they often encounter problems. So, for the Future Recommendations part, the authors should propose more specific development directions for the design of photocrosslinkable biomaterials used for 3D bioprinting that will work in the preparation strategies of low cost scaffold materials strategy in the biomedical engineering field.
Author Response
- We would like to thank you for taking the time to review this manuscript. We have revised the manuscript according to the comments, and some inconsistencies and inaccuracies have been amended as well. Please find the detailed responses below and the corresponding revisions/corrections were highlighted in red in the re-submitted files.
- Comments 1:
- Compared to other crosslinking methods, the authors should explain the advantages of photocrosslinking reactions in 3D bioprinting.
- Response 1:
- We agree it is meaningful to highlight the merits of photocrosslinking reactions in 3D bioprinting. Compared to other physical or chemical crosslinking strategies, photocrosslinking is suitable for 3D bioprinting for its simple, rapid, and precise control over the curing process, which helps build scaffolds with desired structure. (Page 4, line 153-156) In this paper we have illustrated corresponding descriptions, which we wish could offer a general understanding of the advantages of photocrosslinking for 3D bioprinting. In order to avoid repetition, we did not emphasize it again.
- Comments 2:
- In “The major 3D bioprinting technologies” part, Figure 1 consists of four parts. The related descriptions in the text are ambiguous and need to be stated more clearly. Such as Figure 1a, Figure 1b…
- Response 2:
- We are sorry for the paper's inaccurate description and have corrected them as Figure 1a, Figure 1b… for clear understanding. (Page 2 line 85, Page 3 line 103, 119, and 131)
- Comments 3:
- For biomaterials, Low-temperature 3D printing technology should be mentioned.
- Response 3:
- We sincerely appreciate your useful advice for low-temperature 3D printing technology. Compared with conventional 3D printing approaches, there is no doubt that low-temperature 3D printing could maintain the bioactivities of biomaterials like gelatin and alginate. However, since low-temperature 3D printing is frequently performed under much lower temperatures[1], as well as the use of toxical organic solvents like 1,4-dioxane and chloroform[2, 3], which is not quite favorable for cell-laden bioprinting. Therefore, in this paper, we did not pay much attention on the application of biomaterials in low-temperature 3D printing. However, we added the work that uses ColMA for low-temperature 3D bioprinting to illustrate its representative application. (Page 10 line 374-376)
- Comments 4:
- Faced with the dilemma that most 3D bioprinting is still in the laboratory stage, there is an urgent need for printing technologies that can be put into practical applications on the market to prove that bioprinting is not just a "paper talk". In fact, many of those experimental technologies still rely on extrusion-based methods, which are usually functional, but when it comes to printing tissues that heavily rely on arrangement, such as muscles and nerve fibers, they often encounter problems. So, for the Future Recommendations part, the authors should propose more specific development directions for the design of photocrosslinkable biomaterials used for 3D bioprinting that will work in the preparation strategies of low-cost scaffold materials strategy in the biomedical engineering field.
- Response 4:
- We sincerely appreciate your useful advice. In the conclusions and future perspectives parts, we have divided it 4 specific directions for the development of photocrosslinkable biomaterials. An extra part discussion regarding the clinical conversion was added, we wish this could give a better understanding. (Page 25 line 832- Page 27 line 917)
- [1] Q. Zhao, C. Liu, Y. Chang, H. Wu, Y. Hou, S. Wu, M. Guo, Low-Temperature 3D Printing Technology of Poly (Vinyl Alcohol) Matrix Conductive Hydrogel Sensors with Diversified Path Structures and Good Electric Sensing Properties, Sensors (Basel) 23(19) (2023).
- [2] X. Xiao, X. Jiang, S. Yang, Z. Lu, C. Niu, Y. Xu, Z. Huang, Y.J. Kang, L. Feng, Solvent evaporation induced fabrication of porous polycaprolactone scaffold via low-temperature 3D printing for regeneration medicine researches, Polymer 217 (2021).
- [3] T. Sun, J. Wang, H. Huang, X. Liu, J. Zhang, W. Zhang, H. Wang, Z. Li, Low-temperature deposition manufacturing technology: a novel 3D printing method for bone scaffolds, Front Bioeng Biotechnol 11 (2023) 1222102.
Reviewer 2 Report
Comments and Suggestions for Authors
The presented manuscript is a review of photocrosslinkable biomaterials used for 3D bioprinting. Overall, it is thorough, organized, and well-presented. Figures are sufficiently incorporated, and key points of the photocrosslinkable biomaterials are discussed in detail. However, some of the sections need improvement.
(a) The introduction section needs attention and clarity. The paragraph transitions are not coherent.
(b) In the second section (major 3D bioprinting technologies melt electro writing (MEW) is missing.
(c) What are the advantages, disadvantages, and specifically limitations not discussed clearly?
(d) Including a table highlighting biomaterials, bio-inks, and their mechanical properties will help readers understand the materials spectrum and advancements around the biomaterials for tissue engineering.
(e) Discussing bioink compatibility, resolution, and speed will be impactful.
(f) How does the composition (structural and functional) of bioinks for different tissue types (soft and hard tissue) change need to be discussed?
(g) Cytotoxicity needs a specific section as it is one of the huge concerns in bioprinting.
(h) The degradation rate and how it impacts based on the tissue types needs more detailed discussion.
(i) The figures and table are well labeled but some of them are not clear due to overcrowding.
(j) Also, the regulatory challenges and clinical application need to be discussed in detail.
Author Response
- We would like to thank you for taking the time to review this manuscript. We have revised the manuscript according to the comments, and some inconsistencies and inaccuracies have been amended as well. Please find the detailed responses below and the corresponding revisions/corrections were highlighted in red in the re-submitted files.
- Comments 1:
- The introduction section needs attention and clarity. The paragraph transitions are not coherent.
- Response 1:
- We are sorry for incoherent paragraph transitions and we have amended it in the revised paper. We wish this could improve the paragraph transitions. (Page 1 line 42; Page 2 line 62-63)
- Comments 2:
- In the second section (major 3D bioprinting technologies melt electro writing (MEW) is missing.
- Response 2:
- We agree that MEW is a high-resolution 3D printing technique that combines elements of electro-hydrodynamic fiber attraction and melt extrusion. The ability to precisely deposit micro to nanometer strands of biocompatible polymers in a layer-by-layer fashion makes MEW a promising scaffold fabrication method for all kinds of tissue engineering applications. However, the printing process of MEW involves with high-temperature heating to facilitate the melting of thermoplastic materials like PCL, PLLA, et. al[1]. Since the high processing temperature is not suitable for cell-laden-based bioprinting, MEW is mainly used in the fabrication of scaffolds without living cells. Therefore, in this paper, we did not provide more details about MEW for 3D bioprinting.
- Comments 3:
- What are the advantages, disadvantages, and specifically limitations not discussed clearly?
- Response 3:
- We sincerely appreciate your good advice and we agree the advantages, disadvantages, and specifically limitations may not be discussed clear in the paper. In the revised paper we added a table about the advantages, and disadvantages of different biomaterials, we wish this could make a better presentation. (Page 22, line 776-Page 24 line 777)
- Comments 4:
- Including a table highlighting biomaterials, bio-inks, and their mechanical properties will help readers understand the materials spectrum and advancements around the biomaterials for tissue engineering.
- Response 4:
- We really appreciate your valuable advice and we have added a table with details of biomaterials and bioinks within the paper. (Page 22, line 776-Page 24 line 777)
- Comments 5:
- Discussing bioink compatibility, resolution, and speed will be impactful.
- Response 5:
- We agree the discussion over bioink compatibility, resolution, and speed will be impactful. In section 2 of the paper, we have discussed and compared the printing resolution, cell density, and speed of different printing approaches, we think this could make impactful understanding. (Page 2 line 78-Page 3 line 136)
- Comments 6:
- How does the composition (structural and functional) of bioinks for different tissue types (soft and hard tissue) change need to be discussed?
- Response 6:
- We really appreciate your valuable advice and we have added a discussion on the impact of different biomaterials in the application of hard and soft tissues. (Page 24 line 802-Page 25 line 817)
- Comments 7:
- Cytotoxicity needs a specific section as it is one of the huge concerns in bioprinting.
- Response 7:
- We really appreciate your valuable advice and we have added a section in the revised paper. (Page 24 line 777-792)
- Comments 8:
- The degradation rate and how it impacts based on the tissue types needs more detailed discussion.
- Response 8:
- We really appreciate your valuable advice and we have added a section in the revised paper. (Page 24 line 793-801)
- Comments 9:
- The figures and table are well labeled but some of them are not clear due to overcrowding.
- Response 9:
- We really appreciate your valuable advice and we have checked and made slight amendation to the figures and tables to make them clearer.
- Comments 10:
- Also, the regulatory challenges and clinical application need to be discussed in detail.
- Response 10:
- We sincerely appreciate your useful advice. In the conclusions and future perspectives parts, we have divided it 4 specific directions for the development of photocrosslinkable biomaterials. An extra part discussion regarding the clinical conversion was added, we wish this could give a better presentation. (Page 25 line 832- Page 27 line 917)
- [1] S. Loewner, S. Heene, T. Baroth, H. Heymann, F. Cholewa, H. Blume, C. Blume, Recent advances in melt electro writing for tissue engineering for 3D printing of microporous scaffolds for tissue engineering, Front Bioeng Biotechnol 10 (2022) 896719.
Reviewer 3 Report
Comments and Suggestions for Authors
SEE ATTACHED PDF

Author Response
- We would like to thank you for taking the time to review this manuscript. We have revised the manuscript according to the comments, and some inconsistencies and inaccuracies have been amended as well. Please find the detailed responses below and the corresponding revisions/corrections were highlighted in red in the re-submitted files.
- Comments 1:
- Title and Abstract
- The title effectively captures the paper’s scope, focusing on photocrosslinkablе biomaterials for 3D bioprinting. However, the abstract lacks clarity in summarizing thе main findings and kеy contributions of thе papеr. It should morе clеarly highlight thе primary outcomеs and significancе of thе rеviеwеd biomatеrials and tеchniquеs.
- Response 1:
- We sincerely appreciate the useful advice and we have amended the abstract in the revised paper, which we wish could highlight the significance of this review. (Page 1 line 15-18)
- Comments 2:
- Introduction
- Thе introduction providеs a comprеhеnsivе background on thе significancе of 3D bioprinting in tissuе еnginееring and thе rolе of photocrosslinkablе matеrials. Howеvеr, it could bе improvеd by rеducing somе rеpеtitivе contеnt and clеarly stating thе papеr’s objеctivеs in thе final paragraph. Additionally, somе kеy rеfеrеncеs in rеcеnt dеvеlopmеnts arе missing, which would еnhancе thе crеdibility of thе ovеrviеw. Add and draw data from the following:
- 3D Printing in Regenerative Medicine: Technologies and Resources Utilized,
- The Progress in Bioprinting and Its Potential Impact on Health-Related Quality of Life,
- Bio-Inspired Materials: Exhibited Characteristics and Integration Degree in Bio-Printing Operations,
- From Static to Dynamic: Smart Materials Pioneering Additive Manufacturing in Regenerative Medicine,
- Cellular Interaction of Human Skin Cells towards Natural Bioink via 3D Bioprinting Technologies for Chronic Wound: A Comprehensive Review,
- Transforming Object Design and Creation: Biomaterials and Contemporary Manufacturing Leading the Way.
- Response 2:
- We sincerely appreciate your good advice and we have amended the repetitive content and added the references in relevant places, reference 76, 6, 74, 22, 4, 37 respectively. (Page 2 line 74-77)
- Comments 3:
- Structurе and Flow
- Thе structurе follows a logical progrеssion, covеring various printing tеchniquеs and photocrosslinking mеchanisms bеforе moving on to matеrial advancеmеnts. Howеvеr, thе sеctions on bioprinting tеchnologiеs could bе condеnsеd, as thе dеscriptions of еxtrusion, inkjеt, stеrеolithography, and lasеr assistеd mеthods arе еxcеssivеly dеtailеd for a rеviеw contеxt. Strеamlining thеsе sеctions would improvе rеadability and focus.
- Response 3:
- We sincerely appreciate your advice. Different printing technologies have different requirements for bioinks, we believe it is necessary to have a brief introduction of the principles, strengths, weaknesses, and applicability of the four major printing technologies. We wish this could make it clear how to choose the appropriate printing methods for different biomaterials.
- Comments 4:
- Contеnt and Sciеntific Dеpth
- Thе rеviеw providеs an in dеpth look into mеchanisms such as frее radical polymеrization, thiol еnе polymеrization, and rеdox photocrosslinking. Howеvеr, somе arеas nееd grеatеr sciеntific rigor:
- Mеchanisms Sеction: Although mеchanisms arе wеll dеscribеd, somе еxplanations arе ovеrly tеchnical without clеarly linking thеm to thеir rеlеvancе in bioprinting applications.
- Response 4:
- We sincerely appreciate the useful advice and we have added the corresponding information to explain the relationship between photocrosslinking mechanism and bioprinting. (Page 4 line 156-159)
- Comments 5:
- Photoinitiators: Thе sеction discussing photoinitiators is lеngthy and could bеnеfit from a summary tablе comparing thеir strengths, wеaknеssеs, and applicability in bioprinting.
- Response 5:
- We are sorry for missing the information on photoinitiators’ strengths, weaknesses, and applicability in bioprinting. We have added it in Table 1. (Page 8, line 313-314)
- Comments 6:
- Biomatеrials: Whilе thе biomatеrials sеction is comprеhеnsivе, thе prеsеntation is unеvеn, with cеrtain biomatеrials (е.g., gеlatin) rеcеiving morе focus than othеrs without a clеar rationalе.
- Response 6:
- We agree that gelatin has gained more attention than other biomaterial in this paper. However, due to its good physical, chemical, and biological properties, great processability, rich sources, and cheap prices, gelatin (or its derivates) are considered to be one of the most promising biomaterials in the field of 3D bioprinting. Therefore, we offer slightly more details about the current research progress of gelatin in this paper.
- Comments 7:
- Novеlty and Contribution
- Thе rеviеw papеr doеs covеr rеcеnt progrеss in thе fiеld, but it lacks a critical pеrspеctivе on thе limitations of еxisting rеsеarch and how thе discussеd advancеs can addrеss thеsе limitations.
- Response 7:
- We agree it is better to show a critical perspective on the limitations of existing research. We have amended the conclusions and future perspective part and discussed current issues over cell survival, development of the crosslinking system, in vivo bioprinting, and clinical conversion. At the same time, we also mentioned the potential solutions of these limitations. We wish this could make a more in-depth understanding. (Page 25, line 831-Page 27 line 916)
- Comments 8:
- Figurеs and Tablеs
- Figurеs arе gеnеrally wеll dеsignеd, but somе illustrations arе complеx and would bеnеfit from simplification. A figurе summarizing kеy applications for diffеrеnt biomatеrials in 3D bioprinting would providе bеttеr visualization of thе contеnt.
- Response 8:
- We sincerely appreciate the useful advice and we have added a table about the application, strengths, and weakness of the representative biomaterials for light-based 3D bioprinting. (Page 22, line 776-Page 24 line 777)
- Comments 9:
- Conclusions and Futurе Dirеctions
- Thе conclusions section should emphasize thе practical implications and potеntial impact of thеsе advancеmеnts on rеal world applications. Additionally, thе futurе dirеctions arе too gеnеral; it would bе bеnеficial to providе morе spеcific guidancе on еmеrging challеngеs or possiblе rеsеarch pathways.
- Response 9:
- We sincerely appreciate the useful advice, and we have specifically described the current limitations and future development directions of photocrosslinkable biomaterials from four aspects in the Conclusions and Futurе perspectives part.
Round 2
Reviewer 3 Report
Comments and Suggestions for Authors
Conratulations, you did an excellent job, paper is highly recommended for publication!